# Does a smart business environment promote corporate investment? a case study of Hangzhou

**Jing-hua Yin**[1], **Hai-ying Song**[1]*, **Ke-xin Zeng**[2]

**1** School of International Business, Zhejiang International Studies University, Hangzhou, Zhejiang, China,
**2** College of Economics, Hangzhou Dianzi University, Hangzhou, Zhejiang, China

* 5374514@qq.com

## Abstract

As a result of business environment reforms in China's Hangzou, the cost of business has reduced, the confidence of Hangzhou enterprises has survived the COVID-19 outbreak, and foreign investment continues to increase. Nevertheless, Hangzhou's business environment has shortcomings, such as insufficient technology, talent, and intelligent infrastructure. Two unresolved questions persist: (i) Has the smart business environment stimulated corporate investment by reducing system costs and boosting corporate confidence? (ii) How do the commercial climate's shortcomings impact the relationship between the intelligent business environment and business costs/confidence? We examined the impact of a local smart business environment on the corporate investment scale in Hangzhou using factor analysis, cluster analysis, linear regression, and path analyses of data from 297 firm managers. Smart governance improved public administration, financing, and rule of law. The business environment promoted investment by increasing business confidence and decreasing institutional costs. Weak intelligent property protection and legal fairness hindered the positive influence of smart governance on business confidence and system costs. This is the first study combining business environment, smart city, and smart governance concepts to analyze the influence of local smart business environments on business confidence, institutional costs, and investment. Our conclusion on the limitation effect of intelligent business environment on enterprise investment attempts to inspire further research on the intersection of business environments and smart cities. The law of intelligent business environment attracting investment obtained in the context of China, the largest developing country with diversified economic development, is of great significance for other developing countries. Countries can attract investment and promote economic development through intelligent governance. Developing countries should construct smart service platforms, coordinate supervision of public credit, reduce financing constraint, construct a government under the rule of law, improve the quality of land management, and protect intellectual property rights.

**Data Availability Statement:** The data is available at the following: https://doi.org/10.7910/DVN/QO6NGD.

**Funding:** This research was funded by a Project Supported by Scientific Research Fund of Zhejiang

Provincial Education Department, grant number
Y202148108, Recipient J-H Y, Zhejiang Provincial
Soft Science Project, grant number 2020C35035,
Recipient J-H Y, and National Social Science
Foundation of China, grant number 17CJL036,
Recipient K-X Z.

**Competing interests:** The authors have declared
that there are no competing interests.

## Introduction

The German newspaper *Der Spiegel* [1] has commended China's growing economy by comparing it with the global economy, which seems to be almost collapsing due to the COVID-19 outbreak. Smart governance based on digital technologies has contributed substantially to containing the pandemic's spread in China. Hangzhou, a leading smart city in China, has created an online platform for production resumption and used a two-dimensional code for public health management. Within eight days, 200,000 companies applied on the platform to resume production, and over 10 million individuals applied for the Alipay Health Code [2], wherein people sign up through a popular wallet app, Alipay, and are assigned a color code—green, yellow, or red—that indicates their health status.

Information technology [3] and trade policy reform [4] are the two main drivers of Hangzhou's business environment. The local government has the long-term goal to build a citywide credit network, and construction of the Credit Hangzhou network had begun as early as 2002 [5]. Currently, the credit network covers the entire city, and Hangzhou's citizens can check their own "city credit reports" and manage their credit using the Alipay app. The information economy serves as an engine for Hangzhou's economic transformation. The smart economy is developing at a high speed and is opening up the information economy. Further, the Hangzhou Cross-Border E-Commerce Pilot Scheme was set up in 2015 to promote the development of international commerce in China [6]. Its aim was to promote online integration and comprehensive services over three-to-five years, and its main tasks were to establish six systems (for information sharing, financial services, intelligent logistics, e-commerce credit, statistical monitoring, and risk prevention and controls) and two platforms (an online "single window" and an offline "comprehensive park").

Hangzhou's business environment reforms have three main outcomes. First, the cost of doing business is reduced. For instance, City Brain, a smart city intelligence program implemented in Hangzhou, has already connected 96 government departments and 317 information system projects, having an average of more than 120 million collaborative observations per day and resulting in a reduced approval process for investment projects from 10 working days to 9.5 hours [7]. Second, the confidence of Hangzhou enterprises did not collapse due to the COVID-19 outbreak. In the second quarter of 2020, the Hangzhou Small and Micro Entrepreneurs Confidence Index was 115.2, indicating that entrepreneurs are confident in the improvement of the external economic environment [8]. Third, foreign investment in Hangzhou continues to increase. In 2020, Hangzhou utilized foreign capital worth USD 7.2 billion, accounting for 45.6% of the growth in the province, which is 13% higher than the national average growth rate [9].

Nevertheless, Hangzhou's business environment has several shortcomings, such as insufficient technology and talent, lack of intelligent infrastructure, and imbalanced resource distribution and policy supply [10]. Furthermore, two unresolved questions persist: (i) Has the smart business environment stimulated corporate investment by reducing system costs and boosting corporate confidence? (ii) How do the commercial climate's shortcomings impact the relationship between the intelligent business environment and business costs/confidence?

This study thus investigates the impact of the local smart business environment on the scale of corporate investment and explains how this environment promotes business investment by ownership type and research and development (R&D) level. To this end, four hypotheses are tested. First, smart governance can improve the business environment. This proposition has not been fully addressed in the literature. Second, smart environments for government services, financing, and the rule of law can enhance enterprise confidence and reduce institutional costs. Third, enhanced business confidence and lower institutional costs lead to a larger

investment scale. The relationship between commercial climate, institutional costs, and business confidence has long been observed and analyzed, but the impact of a smart business environment on institutional costs and business confidence has not been fully explored.

Business environment studies have described this concept as being comparable worldwide and suggested relevant assessment methods [11,12]. Nevertheless, their theoretical foundations are somewhat weak because, generally, there is an inverse relationship between the intention and denotation of a concept [13]. As some researchers viewed smart governance as a sub-topic of smart cities [14], this issue needs to be researched separately and in more depth.

Overall, as few studies [15,16] exist in this area, this study closes the research gap and is the first to combine the business environment, smart city, and smart governance concepts to analyze the influence of the smart business environment and local smart business on business confidence, institutional costs, and investment. Our findings are unique as they indicate that smart business environments increase business confidence, decrease institutional costs, and then promote investments. In addition, this study presents a firsthand municipal case study from a major developing country, suggesting meaningful policy implications for similar cities in other developing countries.

The results can be generalized to other developing countries. Our conclusions on the limitation effect of intelligent business environment on enterprise investment attempts to inspire other researchers to conduct further research on the intersection of business environments and smart cities. The law of intelligent business environment attracting investment, obtained in the context of China—the largest developing country with diversified economic development—is of great significance for other developing countries. This is particularly significant when the world economy is greatly affected by the COVID-19 outbreak, and countries can attract investment and promote economic development through intelligent governance.

## Literature review

### Business environment

The most influential definition of the business environment comes from the World Bank [12], which highlights the roles of regulatory bodies, administrative systems, and institutions, and focuses on the rule of law (legislation and supervision) for small- and medium-sized enterprises. Based on this, China's official documents define the business environment as institutional factors and conditions for enterprises and other market entities in market economic activities [17]. As this definition focuses on China's business environment reform, it is adopted in this study.

Business environments are too complex to establish under a universally recognized theoretical framework. Therefore, based on previous studies [15,16,18], this study revised the concept of the commercial environment by adding emerging intelligence about digital public services, big-data-based social credit systems, and City Brain.

### Smart city

The unprecedented rate of urban growth makes it imperative to find smarter ways to deal with the challenges accompanying it. Since the "smart city" concept was proposed by International Business Machines Corporation, developed economies have taken the lead in building smart cities around the world. With the vision of improving governance abilities and effectively solving citizens' problems, a smart city is a high-level form of urban informatization and data mobilization [19]. It includes six abstract aspects—smart economy, mobility, life, environment, people, and governance—in a horizontal view [19], as well as three cognitive layers—technical, human, and institutional [20].

However, the practice of constructing smart cities is not satisfactory. For example, there has been insufficient citizen participation in Europe, Canada, and Russia [21–24]; an unsustainable tech-centric approach in China [25]; and immature urban governance in India [26]. As these shortcomings are unresolved, this study explores the leading role of the government in both smart cities and the business environment.

## Smart governance

Governing a smart city involves creating an efficient climate of collaboration within the government and adapting policy-making to new internal and external human cooperation [27]. Smart governance is not a technical issue but a complex process of institutional change with human cooperation as an essential component. However, public participation does not necessarily happen, as vested interests still determine governance around the world [28].

European smart city leadership can be observed mainly in the professional/public-service sector, where local government managers are the main leaders [23], while US smart cities adopt participatory governance structures that emphasize an improvement in public services to attract businesses and development of a collaborative approach to build digital infrastructure [29].

However, smart city governance is more government-driven in China than in Western countries. Here, given the policy and financial support from the central government, local governments play leading roles in fostering and developing smart industries and infrastructure [29].

Compared with smart governance, developing a smart city is more extensive and involves less depth. As the most important sub-criteria in smart city practice, smart governance emphasizes the organizational process to achieve desirable smart city outcomes [18]. Therefore, based on the above-cited previous studies, this study analyzes the relationship between smart governance and the business environment in a smart city.

## Commercial climate of a smart city

Some scholars noticed the imperative impacts of smart city development on the business environment. For example, Porro et al. [15] create a framework for business environments in a smart city, and Blanck et al. [16] propose a smart city indicator system with factors such as public and social service in relation to public service as well as transparent governance in relation to the legal environment. However, as these authors treat smart cities as the target and not the background, their works differ from ours in purpose. Therefore, this study identifies the causal chain of smart governance, business environment, and business investment in a smart city and analyzes the smart business environment and its impacts on investment.

## Theoretical framework

Fig 1 shows our theoretical framework, which is based on Meijer and Bolívar [18].

**Smart governance: Implementation strategies.** The Hangzhou government's vision is to build a smart and innovative city—an East Asian city with a high quality of life—ultimately leading to a happy and harmonious Hangzhou [30]. Therefore, they have issued various local regulations, as well as developed systems and policies to facilitate an information-driven and smart economy and to support entrepreneurship and innovation.

**Smart governance: Outcomes.** The first outcome of smart governance is the shift toward an efficient and transparent system; this implies a sound public service and an efficient administrative environment for doing business. Further, in emerging markets, strengthening of

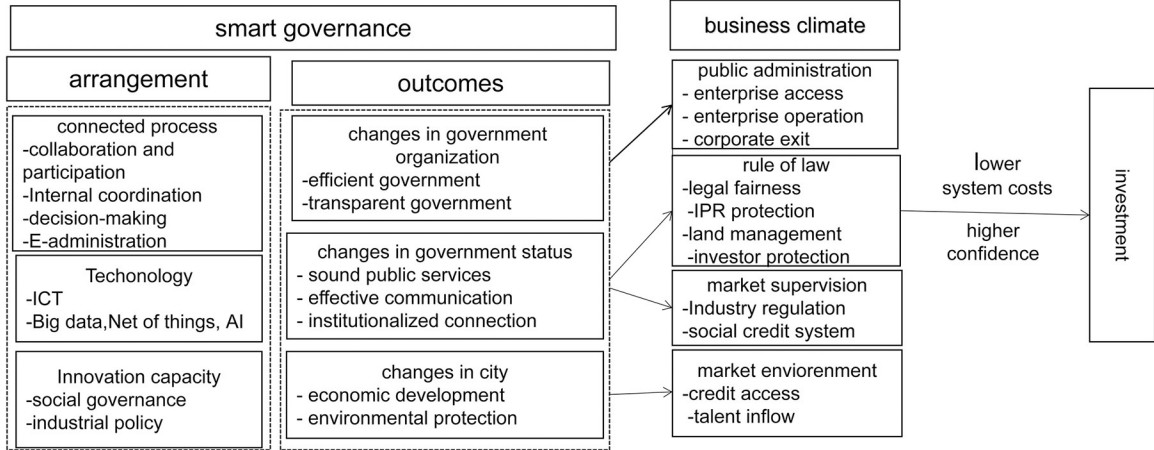

**Fig 1. Theoretical model.** ICT: Information and communications technology; AI: Artificial intelligence; IPR: Intellectual property rights.

institutions creates a more stable business environment, which helps companies focus on long-term innovation rather than short-term competition [31].

The second outcome refers to a change in the relationship between the government and other urban players. The importance of political connections decreases with improvement in the business environment [32]. This change in the government's position is closely related to the legal environment. Governments and enterprises interact more frequently online than off-line and, thus, form institutionalized connections that reduce the entrepreneurs' dependence on personal networks. As a result, enterprises can enjoy better public services, greater business convenience, and lower institutional costs. Meanwhile, governments are more transparent, effective, and less likely to be corrupt. Moreover, the change in the government's position positively influences the financing environment [18]. Smart platforms based on big data can improve market supervision, promote business contract performance, increase corporate profits, and improve external financing environments. Additionally, property rights protection reduces the cost of supervision and prevents the infringement of corporate rights by powerful political groups [33].

The third outcome refers to a change to the city that involves economic development and an improved natural environment. Smart governance can reduce the administrative and management burdens of enterprises, improve corporate efficiency, and activate the business sector [34]. A smart and sustainable city must also ensure that it meets the needs of present and future generations in terms of the economy, society, and environment. Natural environment protection can improve air quality, create a more beautiful and livable environment, and help attract talent and business.

*Benefits of smart governance*: *Business environment optimization through government reforms, collaborative governance, and digital technology.* The business environment consists of the institutional factors and conditions for enterprises and other market entities to participate in market economic activities. Therefore, this study evaluates the business environment in Hangzhou as related to the government (enterprise entry, operation, and exit), legal (property rights protection, judicial fairness, and social credit system), and financing environments as well as other aspects.

First, open data strengthens the government's external supervision and increases citizens' trust, while a one-stop government platform reduces communication costs, improves service

efficiency, and promotes development of the business ecosystem [35]. Second, multi-agent collaborative governance improves the business environment [36]. Similarly, active participation of citizens can improve property rights protection, judicial justice, government efficiency, and democratic decision-making. Third, digital technology improves decision-making and governance capabilities. Smart cities use big data to solve problems through knowledge discovery, knowledge reorganization, inter-organizational cooperation, and post-evaluation [37]. Finally, smart governance improves corporate efficiency and regulatory quality, reduces the administrative and management burden of enterprises, improves their efficiency, activates the commercial sector [34], and improves the fairness and efficiency of supervision at a lower cost.

Therefore, we posit the first proposition as follows:

**Proposition 1:** Smart governance can improve the business environments for government services, financing, and the rule of law.

*Benefits of smart governance*: *Promotion of corporate investment by boosting corporate confidence and reducing institutional costs*. First, an efficient and transparent government environment can increase corporate confidence and reduce system costs [38]. The government's increased market supervision can promote the standardized performance of contracts, increase corporate profits, and improve the external financing environment. The reshaping of government business processes can also improve administrative efficiency, save money and time, and help enterprises seize investment opportunities [39]. Moreover, institutionalized government–enterprise relationships can reduce the entrepreneurs' dependence on personal networks and make them more optimistic toward the business environment [40].

Second, a favorable financing environment can enhance corporate confidence and reduce institutional costs. Financing constraint is caused by information asymmetry in the financial market [41] due to agency problems of the first type (management maximizes its own interests and harms shareholders' interests) and second type (controlling shareholders maximize personal interests and hurt minority shareholders) [42]. Optimizing the business environment and promoting the development of multi-level capital markets can solve the above issues and ease financing constraint by expanding corporate financing channels and increasing financing opportunities [38].

Third, a fair legal environment can enhance corporate confidence and reduce system costs. Judicial justice can effectively determine rules and decrease the development-related uncertainties and transaction costs that enterprises face [43]. Property rights protection can also reduce the cost of supervision and prevent powerful political groups from infringing upon the rights and interests of enterprises. These two can also reduce market expansion costs. Based on the above discussion, we make two propositions:

**Proposition 2:** Smart environments for government services, financing, and the rule of law can reduce institutional costs.

**Proposition 3:** Smart environments for government services, financing, and the rule of law can enhance enterprise confidence.

*Benefits of smart governance*: *Higher business confidence and lower system costs lead to higher investment scales*. Perceived public administration, services, and infrastructural and environmental protection increase citizens' convenience and reduce stress [44]. In the context of a smart city, perceived institutional costs of a company are analogous to the stress perceived by individuals and so is the confidence of a company to individuals' perceived convenience.

Confidence is ultimately a comprehensive judgment made by economic subjects after synthesizing various types of economic information [45]. Business confidence has a fundamental effect on investment as, under specific macroeconomic conditions and low institutional costs, optimistic investors tend to expect high returns and low risks for investment, which are reflected by the higher confidence of enterprises and larger investment volumes [46].

Furthermore, if the business environment is stable and predictable, an individual business will usually expand production, and reduce investment otherwise. Therefore, we derive the fourth proposition as follows:

**Proposition 4:** Enhanced business confidence and lower institutional costs can increase investment.

## Methodology

### Questionnaire design

Our study considers several primary categories related to the commercial environment, including public administration (for business entry, operations, and exit), property rights protection, judicial fairness, access to finance, social credit system, operation cost, and business entry cost. (Due to space considerations, we use "entry cost" instead of "business entry cost" in the following).

The questionnaire respondents were asked to state their company's investment scale in Hangzhou and their confidence in Hangzhou's commercial future. Their satisfaction was assessed on a seven-point Likert-type scale, ranging from 1 (do not agree at all) to 7 (totally agree). Other variables were measured by asking the respondents to evaluate public administration, property protection, and legal fairness.

One key point is how this survey reflects the smartness of Hangzhou using three indicators. First, all questions about public administration, including tax payment convenience, foreign exchange write-off business, convenience of export and import, bankruptcy convenience, intellectual property registration and transfer convenience, entry convenience, and e-commerce pilot zone policy relate to smart city construction.

Second, City Brain uses the Ali Cloud artificial intelligence technology to conduct global real-time analyses of the entire city, while automatically allocating public resources and correcting bugs in the city operation.

Third, the social credit system is based on big data. In 2018, Hangzhou ranked first among 12 demonstration cities in the construction of a social credit system, which has been prioritized by China's National Development and Reform Commission.

### Sampling procedure

The survey was conducted in Hangzhou between September and December 2018, both in person and online. The respondents answered the questions using smartphones by scanning a two-dimensional code or through the survey website, Wen Juan Xing. The sampling process included several stages. First, for accessibility, data were collected in six major industrial parks that housed processing businesses, international e-commerce firms, and technological enterprises. As a result, we collected 82 completed questionnaires. Second, we used an online survey to collect more data, and obtained more than 300 additional completed questionnaires. Third, we read the questionnaires and eliminated those with problematic answers, leaving us with 297 valid responses. Invalid answers included blank answers and multiple submissions from the same Internet protocol address, among others. Fourth, reliability and validity were verified.

Because middle managers and above have a better picture of the organization than frontline employees, the respondents were limited to middle managers or above in Hangzhou enterprises, and they were randomly selected from among 18,000 members of the Wen Juan Xing database. Of these, 6.85% were from Hangzhou and 10.2% were enterprise managers. This website has a membership of 2.6 million, and it randomly invites daily users (over 1 million persons) to be sample members.

**Table 1. Characteristics of sample enterprises.**

| Number of employees | Percentage |
|---|---|
| Under 100 | 41.4 |
| 100–200 | 21.2 |
| 200–500 | 18.2 |
| Above 500 | 19.2 |
| Total | 100 |
| Years of doing business in Hangzhou | Percentage |
| Under 10 | 30 |
| 10–15 | 32.3 |
| 15–20 | 16.8 |
| Above 20 | 20.9 |
| Total | 100 |
| R&D staff percentage | Percentage |
| Under 10 | 68.4 |
| 10–15 | 12.1 |
| 15–20 | 8.4 |
| Above 20 | 11.1 |
| Total | 100 |
| Nature of ownership | Percentage |
| State-owned | 27.3 |
| Foreign | 6.7 |
| Private | 66 |
| Total | 100 |

## Sample background

Table 1 illustrates the sample's constitution. Table 2 provides descriptive statistics for each variable. Table 3 shows descriptive statistics for variables in the estimation models; most are from the factor analysis of the original data and, thus, have 0 as the mean and 1 as the standard deviation. The sample size is 297.

## Methods

This study employed three analysis methods: factor analysis, cluster analysis, and linear regression with backward deletion.

**Table 2. Descriptive statistics.**

| Scope | Variable | Measurement methods | Min | Max |
|---|---|---|---|---|
| Public administration | Public administration | principal component analysis | -2.8826 | 2.3738 |
| Legal environment | Social credit system | Social credit system | 2.0000 | 7.0000 |
| | Property rights protection | principal component analysis | -3.0836 | 1.9263 |
| | Legal quality | principal component analysis | -3.3589 | 1.9334 |
| Market environment | Talent | Talent | 1 | 7 |
| | Financing constraint | principal component analysis | -3.6046 | 2.0797 |
| System cost | Operation cost | principal component analysis | -2.5602 | 2.7603 |
| | Entry cost | principal component analysis | -2.5909 | 2.1080 |
| Infrastructure | Infrastructure | principal component analysis | -3.7902 | 1.7128 |

Resource: Principal component analysis and survey.

**Table 3. Descriptive statistics for the original data.**

| Variable | Index | Mean | Standard deviation |
|---|---|---|---|
| Public administration | Tax payment convenience | 5.24 | 1.305 |
| | Foreign exchange write-off business | 4.82 | 1.248 |
| | Convenience of export and import | 4.88 | 1.196 |
| | Future convenience of export and import | 5.25 | 1.267 |
| | Bankruptcy convenience | 4.52 | 1.274 |
| | IP registration and transfer convenience | 4.66 | 1.192 |
| | Entry convenience | 4.92 | 1.496 |
| | E-commerce pilot zone policy | 5.06 | 1.214 |
| Legal environment | IPR protection | 4.8 | 1.294 |
| | Investor protection | 4.97 | 1.288 |
| | Judicial procedure quality | 4.87 | 1.242 |
| | Land management quality | 4.75 | 1.313 |
| | Social credit system | 5.12 | 1.242 |
| Financing constraint | Financial products | 4.87 | 1.319 |
| | Bank credit line | 4.86 | 1.309 |
| | Interest rate | 4.61 | 1.321 |
| | Credit guarantee agency | 4.86 | 1.304 |
| Entry cost | Registration cost | 4.28 | 1.549 |
| | Registration time | 4.64 | 1.571 |
| Operation cost | Income tax burden | 4.59 | 1.255 |
| | Value added tax burden | 4.16 | 1.402 |
| | Social security premium burden | 4.12 | 1.472 |
| | Time for export and import | 4.31 | 1.399 |
| | Expected future time of export and import | 3.49 | 1.295 |
| | Contract enforcement cost | 4.49 | 1.323 |
| | Legal costs of dispute resolution | 4.41 | 1.257 |
| | Business bankruptcy cost | 4.41 | 1.284 |
| | IPR registration cost | 4.11 | 1.347 |
| Talent | The number of qualified talents | 4.81 | 1.306 |
| Infrastructure | B2B service | 5.17 | 1.25 |
| | Entrepreneurship supporting facility | 5.09 | 1.22 |
| Social credit system | Social credit system | 5.12 | 1.24 |
| City Brain | City Brain | 5.16 | 1.298 |
| Business confidence | Business confidence | 5.31 | 1.301 |

Note: IP: Intellectual property; IPR: IP rights.

Data source: Authors' survey.

**Principal factor analysis.** This study considered five dimensions of the business environment with specified factors. To focus on key issues, we conducted principal factor analysis for each dimension's index. This approach partially eliminated the correlations between variables [47].

First, the value for the public administration environment was obtained from factor analysis of the convenience of enterprise establishment, tax payment, export and import, foreign exchange write-off business, bankruptcy, intellectual property rights (IPR) registration and transfer, and e-commerce pilot zone policy.

Second, the value of the legal environment was obtained from factor analysis of the protection of IPR and investors, quality of judicial procedures, and land management. Furthermore, we separated legal fairness from property rights protection, as the judiciary administration of justice prevents both parties in a market transaction from defaulting, while property rights protection prevents powerful political groups from infringing on enterprises [48].

Third, we separated social credit system from other financing and legal variables, as Hangzhou has implemented a new social credit system based on the Internet and big data, which is jointly supervised by the government and the private sector. The local government has been striving to realize its long-term aim of building a city-wide credit network, and as Hangzhou ranked first among 12 demonstration cities in the construction of a social credit system in 2018 [49], we are interested in evaluating the influence of its credit systems.

Fourth, a financing constraint was generated from the sub-criteria factor analysis. Digital governance improves the legal environment and investor protection and reduces investment risk, thus easing the financing constraint. Furthermore, trust-related factors have a greater impact on the relationship between small and medium-sized enterprises and banks than transaction-related factors [50]. Therefore, by examining this variable, we can see how credit access under digital governance influences investment.

Fifth, the entry cost was obtained from a factor analysis on business registration cost and registration time, while the operational cost was obtained from the burden of income, value added, and social security taxes; time required for export and import; and costs of contract enforcement compliance, legal dispute resolution, bankruptcy, and IPR registration. Huang et al. [51] show that prudent government intervention may damage enterprises' contractual enforcement quality, while access intervention does not. Therefore, we consider the possible influences of enterprise establishment and operation costs on investment.

Finally, our aim is to examine the impact of smart governance on enterprise investment; therefore, we consider B2B intermediary services and entrepreneurship supporting facilities as indicators of urban infrastructure.

**Linear regression with backward deletion.** Accordingly, Model 1 below expresses the relationship between business environment and business confidence postulated in Proposition 3. Models 2 and 3 test the relationship between institutional cost and smart business environment in Propositions 2. Model 4 verifies the influence of business confidence and institutional cost on business investment postulated in Proposition 4. Furthermore, we estimate each model using different sub-samples:

$$
\begin{aligned}
Confidence = a_0 + a_1 * PADM + a_2 * Socredit + a_3 * Citybrain + a_4 * Finconstraint + a_5 \\
* Protection + a_6 * Legalfairness + a_7 * Talent + a_8 * Infrastructure + \varepsilon
\end{aligned} \quad (1)
$$

$$
\begin{aligned}
Operation\ cost \\
= b_0 + b_1 * PADM + b_2 * \text{Socredit} + b_3 * \text{Citybrain} + b_4 * \text{Finconstraint} + b_5 \\
* Protection + b_6 * \text{Legalfairness} + b_7 * Talent + b_8 * Infrastructure + v
\end{aligned} \quad (2)
$$

$$
\begin{aligned}
Entry\ cost = f_0 + f_1 * PADM + f_2 * \text{Socredit} + f_3 * Citybrain + f_4 * \text{Finconstraint} + f_5 \\
* Protection + f_6 * \text{Legalfairness} + f_7 * Talent + f_8 * Infrastructure + u
\end{aligned} \quad (3)
$$

$$
Investment = z_0 + z_1 * Confidence + z_2 * Operation\ cost + z_3 * Entry\ cost + \mu \quad (4)
$$

where Investment is a measure of the investment scale and Confidence refers to business confidence. PADM signifies the public administration environment, and Socredit refers to social credit system. Citybrain indicates the infrastructure of transportation and telecommunication

with smart governance. Finconstraint represents financing constraint, Protection represents property protection, and Legalfairness represents legal fairness. Similarly, Entrycost means entry cost, and Operationcost refers to operational costs. Talent means the qualified talents and Infrastructure represents B2B service and entrepreneurship supporting facility. As shown in Table 4, the values for the public administration environment, legal environment, financing constraint, entry cost, and operational cost are taken from the principal factor analysis. Meanwhile, the values of social credit system and City Brain are taken from the survey data.

**Path analysis.** Path analysis was used to conduct a comprehensive analysis of the interaction between variables [52]. We use path analysis to test Proposition 1 and Propositions 2–4.

First, the path model was determined. According to the theoretical framework, we derived several causal paths: (i) Intelligent governance, business environment (government affairs, law, market, credit), institutional cost, and enterprise confidence all affect enterprise investment. (ii) Institutional costs are affected by smart governance and the business environment. (iii) Business confidence is influenced by smart governance and the business environment. (iv) Smart governance also impacts the business environment.

Second, the regression coefficient was calculated. According to the path model, we investigated the path coefficients of each endogenous variable. These variables were taken as dependent variables, and all the variables related to them were taken as independent variables for multiple regression analysis by the forced entry method. Standardized regression coefficients were taken as path coefficients.

Specifically, to simplify the research question, we only used City Brain to represent smart governance. Previously, we had used City Brain, the smart business environment, and big-data-based social credit system to indicate smart governance, but as the business environment includes smart- and non-smart elements, the social credit system is a part of it.

Third, the path diagram was completed. According to the results of several regressions, the standard coefficient of the output was the path coefficient (direct effect). Then, we conducted error estimation and explained the results.

## Results and discussion

### Linear regression

As shown in Tables 5–8, Proposition 2 is partly verified, as public administration reduces institutional cost, while City Brain increases operation cost but decreases entry cost. Similarly, legal fairness leads to lower operation costs. In addition, infrastructure promotes business confidence, and talent decreases operational costs but increases entry costs. This result implies talent shortages hinder the beneficial functions of a smarter business environment.

Table 5 shows that Proposition 3 holds. Better business environments under smart governance improve business confidence significantly, although social credit system is less significant. Nevertheless, Table 5 presents some unexpected results. Weak IPR protection and the underdeveloped property rights protection regime harms foreign enterprises' business confidence. First, this corresponds with the weak IPR protection we identified in the factor analysis. It is also consistent with Tran [53] in that corruption increases the operating and debt financing costs of enterprises and reduces their risk-taking. In fact, by reducing market costs and improving technical income, a higher IPR protection level in the urban context is conducive for foreigners to purchase more shares from local enterprises or to invest directly [54].

Meanwhile, the current active delisting system of listed companies in China only protects small and medium investors within the framework of the capital market law, but it ignores them as shareholders and contract parties [55]. As for the lack of modern corporate governance structures, insider control dominates listed companies [56].

**Table 4. Factor analysis results.**

| Public administration | Variable | Component |
|---|---|---|
| | Export and import procedure | 0.785 |
| | Tax payment | 0.76 |
| | Foreign exchange write-off business | 0.759 |
| | IP registration and transfer | 0.702 |
| | E-commerce pilot zone policy | 0.657 |
| | Business registration | 0.624 |
| | Business exit | 0.617 |
| | Cumulative percentage | 49.48% |
| **The overall legal environment** | **Variable** | **Component** |
| | Investor protection | 0.833 |
| | Land management quality | 0.825 |
| | Legal procedure quality | 0.824 |
| | IP protection | 0.707 |
| | Cumulative percentage | 72.334 |
| **Legal fairness** | **Variable** | **Component** |
| | Land management quality | 0.888 |
| | Legal procedure quality | 0.888 |
| | Cumulative percentage | 78.896 |
| **Property rights protection** | **Variable** | **Component** |
| | Protection for investors | 0.85 |
| | IP protection | 0.85 |
| | Cumulative percentage | 72.334 |
| **Financing constraint** | **Variable** | **Component** |
| | Credit line supplied by banks | 0.838 |
| | Financial products of banks | 0.828 |
| | Credit guarantee agency | 0.779 |
| | Interest rate of banks | 0.764 |
| | Cumulative percentage | 64.495 |
| **Entry cost** | **Variable** | **Component** |
| | Business registration time | 0.882 |
| | Business registration cost | 0.882 |
| | **Cumulative percentage** | **77.838** |
| **Operation cost** | **Variable** | **Component** |
| | Legal dispute resolution | 0.742 |
| | Contract enforcement compliance | 0.723 |
| | Social security | 0.713 |
| | Value added tax | 0.704 |
| | Bankruptcy | 0.704 |
| | Income tax | 0.658 |
| | Export and import time | 0.653 |
| | IP registration | 0.48 |
| | Cumulative percentage | 45.794 |
| **Infrastructure** | **Variable** | **Component** |
| | B2B service | 0.881 |
| | Entrepreneurship supporting facility | 0.881 |
| | Cumulative percentage | 77.588 |

Note: IP: Intellectual property.

Data sources: Principal factor analysis and the authors' survey.

**Table 5. Estimation results of model (1).**

| Independent variable: Business confidence | Overall | Overall | Private company | Foreign company | State-owned company | Low-tech company | Middle-tech company | High-tech company |
|---|---|---|---|---|---|---|---|---|
| City Brain | 0.119** | 0.1808** | 0.2250** | | | 0.337** | | |
| Public administration | 0.354** | 0.4466** | 0.5770** | | | 0.325** | 0.645** | 0.429** |
| Social credit system | 0.095 | 0.1161** | 0.1210** | 0.769** | | 0.201** | | |
| Financing constraint | 0.161* | 0.1775** | | 0.563** | 0.5490 | | | 0.317** |
| Property rights protection | -0.050 | | | -0.709** | | | | |
| Legal quality | 0.122 | | | | 0.5290** | | 0.252** | 0.291** |
| Talent | -0.010 | | | | | | | |
| Infrastructure | 0.189** | | | | | | | |
| Constant | 4.254** | 3.7782** | 3.4810** | 1.7200** | 5.2840** | 2.4030** | 5.290** | 5.423** |
| Adjusted R$^2$ | 0.391 | 0.3865 | 0.4470 | 0.4980 | 0.3450 | 0.381 | 0.386 | 0.401 |

Note

** $^{means}$ the coefficients are significant at 0.05.

* means the coefficients are significant at 0.10.

However, foreign companies' confidence still promotes investment strongly because the positive influence of social credit system overcomes the negative impacts of poor property rights protection. Western companies are used to doing business under a highly developed social credit system, whereas China's central bank's social credit system only covers about a quarter of the population [57]. Smart credit supervision thus has a significant effect on the business confidence of foreign enterprises through information collection, data integration, and data sharing under a social credit system.

Table 7 shows that, as underdeveloped legal fairness increases, the entry costs for low-tech enterprises increase. There are many comprehensive reasons for this finding. First, local industrial policies favored high-tech enterprises, eliminated traditional and backward production capacity, focused on IPR protection, and cracked down on piracy and infringement. These

**Table 6. Estimation results of model (2).**

| Independent variable: Operation cost | Overall | Overall | Private company | Foreign company | State-owned company | Low-tech company | Middle-tech company | High-tech company |
|---|---|---|---|---|---|---|---|---|
| City Brain | -0.108** | 0.1024** | 0.0810** | | 0.2220** | | | 0.104* |
| Public administration | 0.278** | 0.2828** | 0.2460** | | 0.7240** | 0.333** | 0.251** | |
| Social credit system | -0.04 | | | | 0.1920** | | | |
| Financing constraint | 0.285** | 0.2687** | 0.3200** | | 0.4540** | 0.213** | 0.295** | 0.356** |
| Property rights protection | 0.137** | 0.1307** | | 0.5450** | | | | 0.355** |
| Legal quality | 0.184** | 0.1753** | 0.2660** | | | 0.280* | 0.211** | |
| Talent | 0.005* | | | | | | | |
| Infrastructure | 0.020 | | | | | | | |
| Constant | 0.741** | 0.5290** | 0.3500** | 0.2110** | 2.1950** | 0.102** | 0.102** | 0.510** |
| Adjusted R$^2$ | 0.463 | 0.4671 | 0.4660 | 0.6380 | 0.5270 | 0.542 | 0.380 | 0.371 |

Note

** means the coefficients are significant at 0.05

* means the coefficients are significant at 0.10.

**Table 7. Estimation results of model (3).**

| Independent variable: Entry cost | Overall | Overall | Private company | Foreign company | State-owned company | Low-tech company | Middle-tech company | High-tech company |
|---|---|---|---|---|---|---|---|---|
| City Brain | 0.107* | 0.1180** | 0.1274** | | | 0.318** | 0.342** | 0.475** |
| Public administration | 0.331** | 0.2719** | 0.2641** | | 0.4332** | | | |
| Social credit system | 0.070 | | | | | 0.280** | | |
| Financing constraint | -0.108 | | | | | | | 0.359** |
| Property rights protection | -0.102 | | | | | | 0.255** | |
| Legal quality | -0.133* | 0.1995** | 0.2665** | | | 0.310* | | |
| Talent | -0.082** | | | | | | | |
| Infrastructure | 0.068 | | | | | | | |
| Constant | -0.515 | 0.6095** | 0.6487** | 0.2539 | 0.0620** | 1.481** | 0.033** | 0.046** |
| Adjusted R$^2$ | 0.084 | 0.0739 | 0.0620 | 0.0000 | 0.1970 | 0.143 | 0.065 | 0.074 |

Note

** means the coefficients are significant at 0.05.

policies may hinder low-tech companies without innovative competencies, thus increasing entry costs. Additionally, judicial proceedings involving digital collaborative governance push market entities to decrease contract defaults, but there is still intangible discrimination in the entry of private enterprises [58]. Second, local governments that rely heavily on land taxes or use fees for revenue have strong incentives to increase housing prices and illegal demolition to alleviate debt risks [59,60]. Third, land legislation lacks top-level design [61], and the misallocation of land resources exists in the manufacturing of computer, communications, other electronic equipment, pharmaceuticals, and the information services industry [62]. Finally, urban development is uneven. Specifically, the high-intensity development of Hangzhou is concentrated in the main city, while the suburban and exurban areas are relatively weak and lagging. Furthermore, the environmental problems of the urban fringe are prominent [63].

Table 8 illustrates that Proposition 4 is verified. Higher business confidence and lower entry and operation costs lead to higher investment, while operational cost reduction has a greater influence than entry cost. This result implies that prudent intervention reduction matters more for business investment than access intervention, in accordance with Huang et al. [51].

Additionally, Table 8 shows that a commercial climate with smart governance positively influences investment in private high- and middle-tech companies by boosting confidence and lowering institutional costs. Further, the business environment enhances the investments

**Table 8. Estimation results of model (4).**

| Independent variable investment | Overall | Private company | Foreign company | State-owned company | Low-tech company | Middle-tech company | High-tech company |
|---|---|---|---|---|---|---|---|
| Constant | 3.880 | 3.755 | 3.081 | 4.178 | 3.324 | 3.980 | 4.301 |
| Business confidence | 0.2479 | 0.2860 | 0.3620 | 0.1810 | 0.389 | 0.212 | 0.168 |
| Entry cost | 0.2015 | 0.1550 | | 0.4100 | | 0.224 | 0.263 |
| Operation cost | 0.2603 | 0.3210 | | | 0.424 | 0.170 | 0.385 |
| Adjusted R$^2$ | 0.1966 | 0.219 | 0.1990 | 0.162 | 0.305 | 0.186 | 0.164 |

Note: Except for business confidence of high-tech enterprises and operating cost of low-tech enterprises, all coefficients are significant at 0.05.

in state-owned and low-tech companies by influencing confidence, as well as the enterprise establishment or operation costs. For foreign enterprises, these factors only impact investment by increasing confidence.

As presented in Tables 5–8, Propositions 2–4 hold for most cases, but the situation becomes complex when considering ownership types and technology levels.

**City Brain increases the operational cost of private and state-owned enterprises, including high-tech enterprises.** City Brain increases enterprises' operational costs, the largest of which are for legal dispute resolution, contract enforcement regulation, and social security tax burden, as City Brain has not been applied to the legal and tax systems. This result is similar to the findings of Tan and Taeihagh [64], in that lack of governance frameworks and regulatory guarantees for smart cities hinder their operation in developing countries.

Furthermore, City Brain has been applied to transportation since 2016 and to livelihood areas since the end of 2018. However, the hard infrastructure of Hangzhou in 2018 was not compatible with the potential governance functions under City Brain, being considerably poorer than its soft and intelligent infrastructure [65,66]. Nevertheless, City Brain reduces the entry cost of private enterprises despite the increasing operation costs. This may be due to the private sector's full participation in the project, which has led to some governance power being transferred to the private sector [67].

**Social credit system increases operational costs for state-owned enterprises.** Social credit system based on the Internet and big data does not reduce the operational costs of state-owned enterprises. This is consistent with the conclusions of Horak et al. [68] that *guanxi* is persistent in China, which is a typically a low-trust society. *Guanxi* here refers to China's informal network, which shapes the characteristics of its society [69] and focuses on interpersonal relationships and exchange of favors [70].

First, laws and regulations related to business credit supervision are not perfect. China's enterprise credit supervision system includes an enterprise credit publicity system, credit information sampling system, joint punishment system for intentional debt default, and construction of a credit information publicity platform, among other measures, which need to be further improved [71].

Second, in Hangzhou, the wholesale and retail industries have high risk for credit fraud due to their low operating costs, flexible scale, low enterprise establishment barriers, and uneven product quality [60].

Third, centralized traditional financial supervision is weak compared with decentralized Internet financial supervision [72]. Personal credit supervision mechanisms lack critical basic data and sufficient sharing and integration for the protection of commercial secrets and market competition.

Finally, the information disclosure of state-owned enterprises is not standardized. Entities responsible for the public release of information are not clearly identified, and the boundary between voluntary and compulsory disclosure is fuzzy. Furthermore, the data publication system is underdeveloped and its supervision flawed [73].

**Financing constraint increases high-tech enterprises' entry costs but decreases their operation costs.** According to the enterprise life cycle theory, high-tech enterprises' life cycle can be divided into the seed, initial, growth, and maturity stages. Angel capital and government investment are the main sources of funding for seed-stage enterprises, while start-ups mostly rely on venture capital and government funds [74].

On the one hand, after a long R&D process and huge upfront investment, high-tech enterprises in the seed stage have to face competition to win the favor of angel capitalists and government investment officers, which may increase their market entry time. On the other hand, potential investors can more accurately check the founders' personal and corporate credit

**Table 9. Error estimation of path analysis.**

|  | $R^2$ | $SE = \sqrt{1 - R^2}$ |
|---|---|---|
| **Public administration** | 0.158 | 0.918, |
| **Legal environment** | 0.189 | 0.926 |
| **Financing constraint** | 0.142 | 0.925 |
| **Social credit system** | 0.091 | 0.953, |
| **Institutional cost** | 0.462, | 0.733 |
| **Business confidence** | 0.380 | 0.787 |
| **Enterprise investment** | 0.220, | 0.883 |

through the big data credit supervision platform, which is part of the smart city environment; any contract default may raise further barriers to enterprise establishment.

However, once high-tech enterprises have survived the seed stage, government help will greatly reduce their operating costs. This is consistent with Blanck et al.'s [16] finding that business incubation is correlated with smart city development.

*Path analysis.* Table 9 indicates the error estimation and Table 10 shows the direct and indirect effects. Based on Tables 9 and 10, we can get the path coefficient resolution in Table 11 and the overall influence paths of Fig 2.

Fig 2 shows the overall influence paths among smart governance, business environments, institutional costs, business confidence, and enterprise investments.

As shown in Fig 2 and Table 11, all four propositions hold. First, City Brain has positive direct effects on the business environment (0.402, 0.438, 0.381, 0.306), which confirms Proposition 1. Second, City Brain only impacts institutional costs through the business environment (0.311), verifying Proposition 2. Third, City Brain influences business confidence both directly (0.201) and indirectly though the business environment (0.219). Moreover, the indirect effect is greater than the direct effect, which supports Proposition 3. Fourth, City Brain influences enterprise investments directly (0.212) and indirectly (0.151) by improving the business environment, lowering institutional costs, and enhancing business confidence, which confirms Proposition 4.

Overall, public administration and financing constraint increase investment though system cost and business confidence. Rule of law affects investment mainly through institutional cost, and social credit system promotes investment mainly through business confidence. These influencing paths are consistent with Tables 5 and 6.

**Table 10. Direct and indirect effects.**

|  | Influencing Path | Influencing effects |
|---|---|---|
| **Direct effect** | City Brain → investment | 0.212 |
| **Indirect effect 1** | City Brain → Public administration → Cost → investment | 0.401×0.329×0.214 = 0.028 |
| **Indirect effect 2** | City Brain → Public administration → Business confidence → investment | 0.401×0.306×0.209 = 0.026 |
| **Indirect effect 3** | City Brain → Legal environment → Cost → investment | 0.438×0.198×0.214 = 0.021 |
| **Indirect effect 4** | City Brain → Financing constraint → Cost → investment | 0.381×0.241×0.214 = 0.019 |
| **Indirect effect 5** | City Brain → Financing constraint → Business confidence → investment | 0.381×0.150×0.209 = 0.012 |
| **Indirect effect 6** | City Brain → Social credit system → Business confidence → investment | 0.306×0.128×0.209 = 0.003 |
| **Indirect effect 7** | City Brain → Business confidence → investment | 0.201×0.209 = 0.042 |
| **Total effect** | 0.212+0.028+0.026+0.021+0.019+0.012+0.003+0.042 | 0.363 |

**Table 11. Path coefficient resolution.**

| Causal variable | Outcome variables | Direct influence | Indirect influence | Total influence |
|---|---|---|---|---|
| City Brain | Public administration | 0.402 | | 0.402 |
| | Legal environment | 0.438 | | 0.438 |
| | Financing constraint | 0.381 | | 0.381 |
| | Social credit system | 0.306 | | 0.306 |
| | Institutional costs | | 0.311 | 0.311 |
| | Business confidence | 0.201 | 0.219 | 0.420 |
| | Enterprise investment | 0.212 | 0.151 | 0.363 |

Data source: Path analysis and the authors' survey.

## Conclusions

This study's findings can be summarized as follows. First, smart governance improves the business environment for public administration, financing, and the rule of law. Second, under smart governance, the business environment promotes investment by enhancing business confidence and decreasing stress.

The results can be generalized to other developing countries. First, our conclusion on the limitation effect of intelligent business environment on enterprise investment can inspire other scholars to conduct further research on the intersection of business environments and smart cities. Second, the rules of an intelligent business environment in promoting corporate investment are still in line with the reality of other countries. As the largest developing country in the world, China's economic development is particularly diversified. The law of intelligent business environment attracting investment, obtained in China's context, is of great significance for other developing countries. Especially now, when the world economic environment is greatly affected by uncertainty, countries can attract investment and promote economic development through intelligent governance.

However, the case study is limited as it focuses on one Chinese city in 2018. Future research should provide empirical evidence on the impacts of smart governance on the business environment and expand the survey period and scope to produce more reliable, generalizable, and

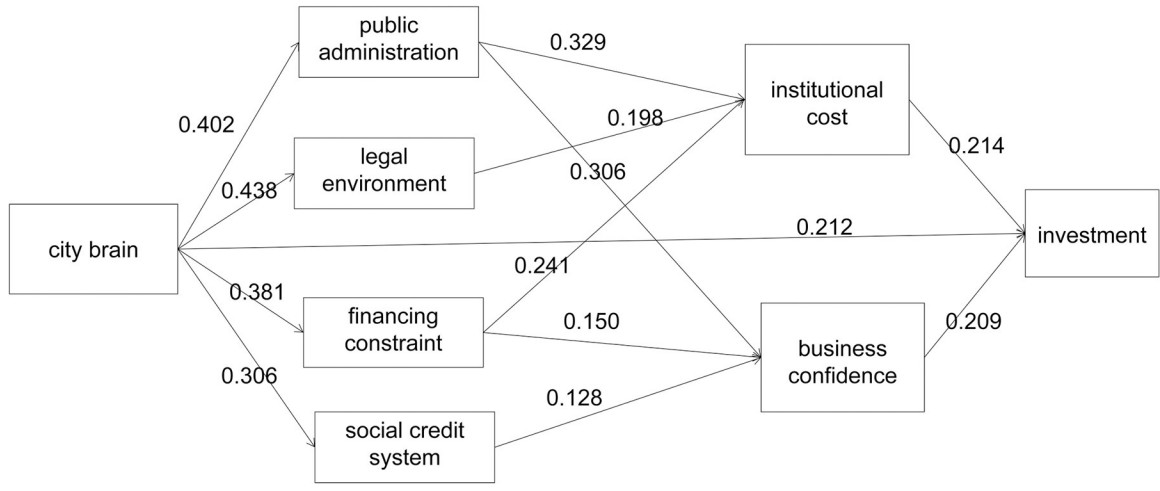

**Fig 2. Path analysis.**

relevant findings. Nevertheless, the findings have meaningful policy implications. To increase investment in the context of smart cities, developing countries should make the following reforms simultaneously. Some of these have been identified by other researchers too, but earlier recommendations are not specific or targeted enough for developing countries.

The first aspect is the construction of a smart service platform, and the recommendations are as follows. (i) There is a need to put forward a specific action plan for the construction of smart cities, accelerate the application of smart service platforms in key government departments, and advance digital government reform in depth. (ii) Considering the original citizens, individuals, and experts participating in politics, a business environment governance community should be cultivated with corporate organizations, industry associations, non-profit organizations, and business-to-business intermediary service organizations constituting the main body. (iii) There is a need to facilitate communication between the government and enterprises through digital channels, create a new type of government–business relationship, and foster a good-governance environment.

The second aspect is the coordinated supervision of public credit. (i) Governments should crack down on platforms that illegally absorb public deposits or engage in fraudulent fundraising and other illegal and criminal activities, as well as encourage Internet finance companies to build core innovation capabilities. (ii) The wholesale and retail industry and other service industries should be regarded as key areas for preventing credit fraud and improving supervision. (iii) There is a need to use emerging technologies such as blockchain, Internet of Things, and big data to create an improved, decentralized, distributed, and real-time financial supervision mechanism for Internet finance companies. (iv) The connection between the central bank's credit system and third-party big data credit supervision systems should be enhanced, and data sharing and integration should be promoted. (v) There is a need to disclose the credit of government agencies step by step, including sensitive information, to eliminate conflicts and duplication of legislation. Additionally, the information disclosure rules of enterprises should be standardized, their information disclosure subjects clarified, voluntary and mandatory disclosure distinguished, the data disclosure system refined, and the level of supervision improved.

The third aspect is to reduce financing constraint. (i) To this end, there is a need to increase the importance and support of technology-based small and micro enterprises and increase the ease and usefulness of their access to technology services; conduct industry research and establish a certification system for high-tech enterprises along with an intangible asset evaluation system; and reduce the financing constraint of micro enterprises by providing policy support. (ii) It is also important to promote breakthroughs and innovations in financing risk mitigation technologies, reduce banks' reliance on guarantees or collateral, and cut off the formation of excessively long guarantee chains. (iii) Government departments and returnee entrepreneurs can communicate effectively through multiple channels, such as digital channels.

The fourth aspect is the construction of a government under the rule of law. (i) Administrative irresolution should be improved and the effectiveness of higher-level agencies' relief of the rights of administrative counterparts and supervision of lower-level agencies should be strengthened. (ii) There is also the need to reduce government debt risks and funding dependence on land taxes and fees, increase the degree of governance in accordance with the law during any demolition process, and severely crack down on cases of infringement of citizens' property rights.

The fifth aspect is the quality of land management. (i) Under this aspect, there is a need to pay attention to the top-level design of land use and eliminate inconsistencies between legislation and practice. (ii) Functions of the main city should be shifted away from the urban core; the radiation effect—spillover of vital economic activity into suburbs and exurbs of the main

city—should be improved; the development of the suburbs should be sped up; and the environmental problems in the urban fringe should be paid more attention. (iii) The land mismatch between manufacturing and service industries should also be reduced and resource allocation effectiveness improved.

The final aspect concerns IPR protection. (i) Here, there is the need to pay attention to the identities of small and medium investors as shareholders of listed companies and parties to sales contracts, improve financial supervision and regulation to fully protect the investors' interests, accelerate the establishment of a modern corporate governance structure for enterprises with a board of directors at the core, establish institutional protection for shareholders' rights and interests, and reduce insider control. (ii) It is important to improve the quality of good innovation by administrative protection and property rights protection, rather than just focus on quantity. (iii) IP infringement cases in the e-commerce sector should be a key area of prevention and control. (iv) There is the need to ensure that the promotion of inter-city IPR protection is integrated, judicial trial standards for IP cases are unified, and an administrative coordination protection is implemented in practice.

## Supporting information

**S1 Fig. Theoretical model.** Fig 1 shows our theoretical framework.
(PNG)

**S2 Fig. Path analysis.** Fig 2 shows the overall influence paths among smart governance, business environments, institutional costs, business confidence, and enterprise investments.
(PNG)

## Author Contributions

**Conceptualization:** Jing-hua Yin, Ke-xin Zeng.

**Data curation:** Jing-hua Yin.

**Formal analysis:** Jing-hua Yin.

**Funding acquisition:** Jing-hua Yin.

**Investigation:** Jing-hua Yin.

**Methodology:** Jing-hua Yin, Hai-ying Song.

**Resources:** Jing-hua Yin.

**Writing – original draft:** Jing-hua Yin.

**Writing – review & editing:** Jing-hua Yin.

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
