## [Decision Letter · Decision Letter 0]

21 Mar 2022

PONE-D-21-22763Does a smart business environment promote corporate investment? A case study of HangzhouPLOS ONE

Dear Dr. Yin,

Thank you for submitting your manuscript to PLOS ONE. After careful consideration, we feel that it has merit but does not fully meet PLOS ONE’s publication criteria as it currently stands. Therefore, we invite you to submit a revised version of the manuscript that addresses the points raised during the review process. The reviewers have raised concerns regarding the scope, implications, contributions and presentation of your work. Please address these considering the PLOS ONE’s publication criteria.

We look forward to receiving your revised manuscript.

Kind regards,

Rashid Mehmood, PhD

Academic Editor

PLOS ONE

Journal Requirements:

4. Please amend the manuscript submission data (via Edit Submission) to include authors H.-Y. Song, and K.-X. Zeng.

Reviewers' comments:

Reviewer's Responses to Questions

**Comments to the Author**

1. Is the manuscript technically sound, and do the data support the conclusions?

Reviewer #1: No

Reviewer #2: Yes

2. Has the statistical analysis been performed appropriately and rigorously? 

Reviewer #1: No

Reviewer #2: Yes

3. Have the authors made all data underlying the findings in their manuscript fully available?

Reviewer #1: No

Reviewer #2: No

4. Is the manuscript presented in an intelligible fashion and written in standard English?

Reviewer #1: No

Reviewer #2: Yes

5. Review Comments to the Author

Reviewer #1: The authors built an approach to studying the impact of the smart business environment in Hangzhou, China on the corporate investment scale. The reforms of Hangzhou’s business environment produced three main outcomes, which included: reducing the cost of doing business, continuing the confidence of Hangzhou enterprises with the COVID-19 outbreak, and increasing foreign investment in Hangzhou. However, Hangzhou’s commerce environment has a few deficiencies, such as a lack of intelligent infrastructure and insufficient technology.

The methodology of the study was built based on the questionnaire method which included different categories for the commercial environment. Then, the study used three types of analysis: linear regression analysis, factor analysis, and path analysis on data. The results of this study show that smart governance enhances the improvement of the business environment for public administration, financing, and the rule of law. Also, the business environment in smart governance promotes investment by supporting business confidence and reducing stress.

The work is useful but the paper lacks a clear structure. The textual and graphical presentation is poor. Some equations are not properly typed.

Reviewer #2: The authors examine in this paper the impact of a smart local business environment on the corporate investment scale in Hangzhou, China, using factor, linear regression, and path analyses on data from firm managers. The authors report important findings including that respondents indicated that smart governance improved public administration, financing, and rule of law.

Smart governance improves the business environment for public administration, financing, and the rule of law. The business environment promotes investment by enhancing business confidence and decreasing stress. Therefore, the authors highlight five aspects for developing countries that should make reforms in order to increase investment in their smart city initiatives. These aspects start with the construction of a smart service platform, coordinated supervision of public credit, reduce financing constraints, construction of a government under the rule of law, and quality of land management and IPR protection.

The authors have made a good effort to combine business environment and smart city concepts. It is a good paper and can be accepted subjected to major modifications.

It would be helpful if further analysis included other variables such as infrastructure and labor quality, which are important factors.

The case study is limited as it focuses on one Chinese city in 2018. Although the authors acknowledge that their study is limited to a small number of developing countries, they do not discuss whether the results can be generalized to other developing countries. (line 101)

The study did not include any discussion of climate change or environmental factors which are increasingly becoming a concern for business decision-makers around the world.

Another concern is the quality of the figures and tabular results. I suggest the authors improve the presentation of the tabular results and consider presenting these using figures/plots. The figure quality should be improved.

Page-17: lines 363-369 some missing formula characters.

6. PLOS authors have the option to publish the peer review history of their article (what does this mean?). If published, this will include your full peer review and any attached files.

Reviewer #1: No

Reviewer #2: No

---

## [Author Response · Author response to Decision Letter 0]

1 Apr 2022

Response letter

Emily Chenette

Editor-in-Chief

PLOS ONE

Dear Editor:

I, along with my co-authors, would like to re-submit the attached manuscript entitled “Does a smart business environment promote corporate investment? A case study of Hangzhou.” The manuscript ID is PONE-D-21-22763.

Thank you for giving me the opportunity to submit the revised version of my manuscript. I appreciate the time and effort that you have dedicated to providing feedback on my manuscript and I am grateful for the insightful comments, which allowed me to make valuable improvements to the paper. 

I have incorporated most of the suggestions made by you and the reviewers. Below, please find the point-by-point responses to the comments and concerns.

I hope these changes fully address the reviewers’ concerns and make the paper suitable for publication in your journal.

Thank you for your consideration. I look forward to hearing from you.

Sincerely,

H.-Y. Song

Zhejiang International Studies University

Email: 5374514@qq.com

Reviewer #1: 

Comment 1 

1.The manuscript is not technically sound, and the data doesn’t support the conclusions.

Response:

Thank you for the comment. We are sorry but we do not agree with your opinion.

We examined the impact of a smart local business environment on the corporate investment scale in Hangzhou, China, using survey data from 297 firm managers. We use factor analysis and linear regression to verify Propositions 2-4, and adopt path analyses to prove Proposition 1.

Four hypotheses are tested. First, smart governance can improve the business environment. This proposition has not been fully addressed in the literature. Second, smart environments for government services, financing, and the rule of law can enhance enterprise confidence and reduce institutional costs. Third, enhanced business confidence and lower institutional costs lead to a larger investment scale. The relationship between commercial climate, institutional costs, and business confidence has long been observed and analyzed, but the impact of a smart business environment on institutional costs and business confidence has not been fully explored.

Comment 2

The statistical analysis hasn’t been performed appropriately and rigorously.

Response:

Thank you for the comment. We are sorry, but we do not agree with your opinion.

First of all, we designed a questionnaire based on a large amount of domestic and foreign literature.

Second, we strictly followed the scientific procedure of random sampling to ensure the randomness of sampling.

Third, we strictly tested the questionnaire data to ensure the reliability and validity of the questionnaire results.

Fourth, we designed the empirical estimation model based on the scientific theoretical framework.

Fifth, we carefully explained the estimated results of the model and compared them with the conclusions of a large number of relevant studies in China and abroad to enhance the credibility of our research results.

Comment 3

The authors haven’t made all data underlying the findings in their manuscript fully available.

Response:

Thank you for the comment. This time we supply all the survey data as supporting materials.

The revisions are in supporting materials.

Comment 4

the manuscript isn’t presented in an intelligible fashion and written in standard English.

Response:

Thank you for the comment. The manuscript has been proofread again by the professional editing agency, Editage.

Comment 5 

The work is useful but the paper lacks a clear structure. 

Response:

Thank you for the comment. We have carried out the following revision to make the structure clearer. 

First, we integrate the discussion of linear regression into the results of linear regression. Second, the format is revised according to the journal’s requirements. Third, 22 tables have been revised to only 11 tables to make the structure of the article clearer.

Comment 6 

The textual and graphical presentation is poor. 

Response:

Thank you for the comment. We have reduced 22 tables to 11 tables and redrawn the 2 figures at a higher resolution. In addition, the manuscript has undergone proofreading by the professional editing agency, Editage.

Comment 7

Some equations are not properly typed.

Response:

Thank you for the comment. We have revised the equations according to the journal’s requirement.

Reviewer #2: 

Comment 1

The manuscript is technically sound, and the data supports the conclusions. 

The statistical analysis has been performed appropriately and rigorously.

The manuscript is presented in an intelligible fashion and written in standard English.

Response:

Thank you very much for your appreciation

Comment 2

The authors haven’t made all data underlying the findings in their manuscript fully available.

Response:

Thank you for the comment. This time we supply all the survey data as supporting materials.

Comment 3

The authors have made a good effort to combine business environment and smart city concepts. It is a good paper and can be accepted subjected to major modifications.

Response:

Thank you for the comment. We have improved the scope, implications, contributions, and presentation.

Comment 4

It would be helpful if further analysis included other variables such as infrastructure and labor quality, which are important factors.

Response:

Thank you for your advice. We agree with your opinion.

Our aim is examining the impact of smart governance on enterprise investment; therefore, we have added B2B intermediary services and entrepreneurship supporting facilities as indicators of urban infrastructure. We have also added talent (qualified numbers of talents) into our regression. The results show that infrastructure promotes business confidence, and talent decreases operational costs but increases entry costs. This result implies talent shortages hinder the beneficial functions of a smarter business environment.

Comment 5

The case study is limited as it focuses on one Chinese city in 2018. Although the authors acknowledge that their study is limited to a small number of developing countries, they do not discuss whether the results can be generalized to other developing countries. (line 101)

Response:

Thank you for the comment. We agree with you and have added the discussion of generalization of the results as follows.

The results can be generalized to other developing countries. First, our conclusion on the limitation effect of intelligent business environment on enterprise investment can inspire other scholars to conduct further research on the intersection of business environments and smart cities. Second, the rules of an intelligent business environment in promoting corporate investment are still in line with the reality of other countries. As the largest developing country in the world, China's economic development is particularly diversified. The law of intelligent business environment attracting investment, obtained in China’s context, is of great significance for other developing countries. Especially now, when the world economic environment is greatly impacted by uncertainty, countries can attract investment and promote economic development through intelligent governance. 

Comment 6

The study did not include any discussion of climate change or environmental factors which are increasingly becoming a concern for business decision-makers around the world.

Response:

Thank you for the comments. Our study lasted only for one year and was a cross-sectional study; therefore, it was difficult to reflect the impact of intelligent governance on the improvement of natural environment or climate change. However, that does not mean that it has no such effect. We will focus on this topic in our future research.

Comment 7

Another concern is the quality of the figures and tabular results. I suggest the authors improve the presentation of the tabular results and consider presenting these using figures/plots. The figure quality should be improved.

Response:

Thank you for the comment. We have reduced 22 tables to 11 tables and redrawn the 2 figures at a higher resolution. In addition, the manuscript has been proofread again by the professional editing agency, Editage.

Comment 8

Page-17: lines 363-369 some missing formula characters.

Response:

Thank you for the comment. We have added the formula characters.

---

## [Decision Letter · Decision Letter 1]

16 May 2022

Does a smart business environment promote corporate investment? A case study of Hangzhou

PONE-D-21-22763R1

Dear Dr. Song,

We’re pleased to inform you that your manuscript has been judged scientifically suitable for publication and will be formally accepted for publication once it meets all outstanding technical requirements.

Kind regards,

Rashid Mehmood, PhD

Academic Editor

PLOS ONE

Additional Editor Comments (optional):

Reviewers' comments:

Reviewer's Responses to Questions

**Comments to the Author**

1. If the authors have adequately addressed your comments raised in a previous round of review and you feel that this manuscript is now acceptable for publication, you may indicate that here to bypass the “Comments to the Author” section, enter your conflict of interest statement in the “Confidential to Editor” section, and submit your "Accept" recommendation.

Reviewer #1: All comments have been addressed

Reviewer #2: All comments have been addressed

2. Is the manuscript technically sound, and do the data support the conclusions?

Reviewer #1: Yes

Reviewer #2: Yes

3. Has the statistical analysis been performed appropriately and rigorously? 

Reviewer #1: Yes

Reviewer #2: Yes

4. Have the authors made all data underlying the findings in their manuscript fully available?

Reviewer #1: Yes

Reviewer #2: No

5. Is the manuscript presented in an intelligible fashion and written in standard English?

Reviewer #1: Yes

Reviewer #2: Yes

6. Review Comments to the Author

Reviewer #1: The authors built an approach to studying the impact of the smart business environment in Hangzhou, China on the corporate investment scale. The authors have addressed my comments and have improved the paper.

Reviewer #2: The authors have addressed my concerns. This is a good paper and will make a good addition to the journal.

7. PLOS authors have the option to publish the peer review history of their article (what does this mean?). If published, this will include your full peer review and any attached files.

Reviewer #1: No

Reviewer #2: No

---

## [Editor Report · Acceptance letter]

24 Jun 2022

PONE-D-21-22763R1 

Does a smart business environment promote corporate investment? A case study of Hangzhou 

Dear Dr. Song:

I'm pleased to inform you that your manuscript has been deemed suitable for publication in PLOS ONE. Congratulations! Your manuscript is now with our production department. 

Kind regards, 

on behalf of

Dr. Rashid Mehmood 

Academic Editor

PLOS ONE